META-RESEARCH ARTICLE

# Ensuring the quality and specificity of preregistrations

**Marjan Bakker**[1‡]*, **Coosje L. S. Veldkamp**[2‡], **Marcel A. L. M. van Assen**[1,3], **Elise A. V. Crompvoets**[1,4], **How Hwee Ong**[5], **Brian A. Nosek**[6,7], **Courtney K. Soderberg**[6], **David Mellor**[6], **Jelte M. Wicherts**[1]

**1** Department of Methodology and Statistics, Tilburg University, Tilburg, the Netherlands, **2** Faculty of Social Sciences, Utrecht University, Utrecht, the Netherlands, **3** Department of Sociology, Utrecht University, Utrecht, the Netherlands, **4** Cito Institute for Educational Measurement, Arnhem, the Netherlands, **5** Department of Social Psychology, Tilburg University, Tilburg, the Netherlands, **6** Center for Open Science, Charlottesville, Virginia, United States of America, **7** Department of Psychology, University of Virginia, Virginia, United States of America

‡ MB and CLSV are co-first authors on this work.
* m.bakker_1@uvt.nl

**Data Availability Statement:** The data and materials for this study are available at https://osf.io/fgc9k/ and the study was preregistered and is available at https://osf.io/k94ve/. An earlier version

## Abstract

Researchers face many, often seemingly arbitrary, choices in formulating hypotheses, designing protocols, collecting data, analyzing data, and reporting results. Opportunistic use of "researcher degrees of freedom" aimed at obtaining statistical significance increases the likelihood of obtaining and publishing false-positive results and overestimated effect sizes. Preregistration is a mechanism for reducing such degrees of freedom by specifying designs and analysis plans before observing the research outcomes. The effectiveness of preregistration may depend, in part, on whether the process facilitates sufficiently specific articulation of such plans. In this preregistered study, we compared 2 formats of preregistration available on the OSF: Standard Pre-Data Collection Registration and Prereg Challenge Registration (now called "OSF Preregistration," http://osf.io/prereg/). The Prereg Challenge format was a "structured" workflow with detailed instructions and an independent review to confirm completeness; the "Standard" format was "unstructured" with minimal direct guidance to give researchers flexibility for what to prespecify. Results of comparing random samples of 53 preregistrations from each format indicate that the "structured" format restricted the opportunistic use of researcher degrees of freedom better (Cliff's Delta = 0.49) than the "unstructured" format, but neither eliminated all researcher degrees of freedom. We also observed very low concordance among coders about the number of hypotheses (14%), indicating that they are often not clearly stated. We conclude that effective preregistration is challenging, and registration formats that provide effective guidance may improve the quality of research.

## Introduction

The scientific method is not a static process or set of techniques, but rather an evolving constellation of practices for formulating hypotheses, making observations, gathering data about

of this manuscript appeared as Chapter 6 of the preprint of the doctoral thesis of the first author (DOI 10.31234/osf.io/g8cjq). The preprint of the current version of this manuscript (DOI 10.31234/osf.io/cdgyh) is available at https://psyarxiv.com/cdgyh.

**Funding:** The research of JW was supported by a Consolidator Grant 726361 (IMPROVE) from the European Research Council (ERC; https://erc.europa.eu/). DM, CS, and BN. were supported by grants from Arnold Ventures (https://www.arnoldventures.org/), Templeton World Charity Foundation (https://www.templetonworldcharity.org/), Templeton Religion Trust (https://templetonreligiontrust.org/), and John Templeton Foundation (https://www.templeton.org/) to BN. The funders had no role in study design, data collection and analysis, decision to publish, or preparation of the manuscript.

**Competing interests:** I have read the journal's policy and the authors of this manuscript have the following competing interests: DM, CS, and BN are employees of the Center for Open Science (COS), a non-profit technology and culture change company with a mission to increase openness, integrity, and reproducibility of research. COS operates the OSF and the Prereg Challenge.

testable predictions, and developing general theories. Such practices include random assignment to treatment conditions, statistical techniques for controlling confounding influences, standards of practice for making statistical inferences (e.g., $p < 0.05$), and transparent reporting of methods and results. Progress in science is marked both by creation of knowledge and improving methodology. One such improvement that is gaining popularity is preregistration [1,2] with particularly rapid adoption in social and behavioral sciences like psychology [3].

The key features of preregistration are (1) a priori specification of the research design and analysis plan; (2) posting the plan in discoverable repositories prior to observing the outcomes of the study; and (3) reporting all of the planned analyses. Preregistration is, therefore, comparable to prospective registration of clinical trials [1,4,5], although with an added focus on registering planned analyses. Specification of the design and the analysis plan before observing the outcomes prevents the outcomes from affecting design and analysis decisions [6–9]. Without preregistration, this problem is prone to occur because of ordinary confirmation, hindsight, and outcome biases that affect human reasoning [10,11]. Reporting all of the planned analyses enables accurate statistical inferences such as avoiding the inflation of false positives in null hypothesis significance testing based on selective reporting of multiple alternative analytic results. And, posting the preregistration and the outcomes in independent, discoverable repositories ensures the discoverability of all research conducted on a topic rather than just the research that was ultimately published in a journal. This reduces the deleterious effects of publication bias on the credibility of the evidence base in the available literature [1].

Preregistration clarifies the distinction between planned and unplanned analyses, often corresponding to confirmatory, or hypothesis-testing research, and exploratory, or hypothesis-generating research. Clearly, distinguishing these 2 modes of research is vitally important for maintaining the validity of statistical inferences in confirmatory analysis and for avoiding mistaking the generation of a hypothesis in exploratory analysis as the testing of a hypothesis in confirmatory analysis [1,2]. Unplanned, exploratory analyses are often interactively influenced by what is observed in the data. Such data-contingent analyses increase the likelihood of false inference and exaggerate effect sizes. Failure to identify them as unplanned or exploratory can therefore reduce the credibility of the findings [12–16]. Note that "unplanned" and "exploratory" are not redundant. One can plan, and preregister, exploratory analyses to retain confidence in statistical inferences when there is little basis for articulating a priori hypotheses.

Even when researchers have hypotheses or plans for analysis, they may fail to specify them clearly or completely enough to eliminate the possibility of making data-contingent decisions that would reduce the credibility of the analysis. For example, a researcher might forget to specify rules for excluding observations and only after observing the data recognize that a choice needs to be made. This creates "researcher degrees of freedom" ([17, 18]; see Box 1). If

## Box 1. Researcher degrees of freedom

Analyses of data involve many (often arbitrary) choices that have to be made during data analysis [30]. Researchers could use these choices opportunistically when confronted with an (undesired) nonsignificant result [14,18,31–33]. This use may result in statistically significant findings after all but might yield overestimated effect sizes and inflated rates of false positives. The opportunistic use of the so-called "researcher degrees of freedom" [18,33] is often denoted "p-hacking." This practice constitutes a large problem because its occurrence is estimated to be high [17,34–39].

data-contingent choices have a material impact on the findings, then it is difficult to know if the researcher was intentionally or unintentionally influenced by the observed outcomes in making those decisions, thereby reducing confidence in the findings. Preregistrations must be "specific," "precise," and "exhaustive" [18]. "Specific" means that the preregistration is detailed in its description of all phases of the research process from the design of the study to what will be reported in the manuscript. "Precise" means that each aspect of the research plan is open to only 1 interpretation. "Exhaustive" means that each aspect of the preregistered research plan explicitly excludes the possibility of deviations from the preregistered research plan. For example, a description like "we will use the Rosenberg self-esteem scale (RSES)" leaves ample room for a decision to select a subset of items or not and to construct the composite score in a manner that yields the most favorable effects. A specific, precise, and exhaustive description would include the protocol to administer the items, the scoring of the items, and the procedure to construct the composite score from the items. This includes specifying how deviating individual items, incorrect values, and missing values will be handled and explicitly clarifying that no other procedure(s) will be used for measuring the dependent variable.

The potential value of preregistration has been known for a long time [12] and has become common for clinical trials [4] with inconsistent emphasis on registration before the study and pre-specification of analysis plans [19]. The practice is now gaining popularity in other fields, particularly the social and behavioral sciences like psychology [3,20]. For example, the number of preregistrations at OSF has approximately doubled yearly with 38 in 2012 to 36,675 by the end of 2019 (http://osf.io/registries). Accompanying the availability of infrastructure to support preregistration is a variety of formats of what should be specified in a preregistration [21–23]. These formats range from ones that offer hardly any instructions to others with instructions to provide a high level of detail about many aspects of the study. At the start of the current study, there were 3 primary preregistration formats at OSF: "Open-ended Registrations," "Standard Pre-Data Collection Registrations," and "Prereg Challenge Registrations."

"Open-ended Registrations" are the most unstructured format in which researchers are only asked "to provide a narrative summary of their project." "Standard Pre-Data Collection Registrations" are similar and ask researchers to indicate whether they have already collected or looked at the data before composing the preregistration. "Prereg Challenge Registrations" (now called "OSF Preregistrations," http://osf.io/prereg/) are the most structured format with 26 questions and instructions to provide substantial detail in answering the questions. The questions pertain to general information about the study (title, authors, research questions, and hypotheses), the sampling plan (whether existing data are used, explanation of existing data, data collection procedure, sample size, sample size rationale, and stopping rule), the variables (manipulated variables, measured variables, and indices), the design plan (study type, blinding, study design, and randomization), the analysis plan (statistical models, transformations, follow-up analyses, inference criteria, data exclusion, missing data, and (optional) exploratory analyses), and the scripts that will be used (optional). This format was developed for and used in the "Preregistration Challenge" (or "Prereg Challenge"), a competition held by the Center for Open Science (COS) from 2015 to 2018 to promote experience and education with preregistration. To be eligible for 1 of the 1,000 prizes of US$1,000, participants had to submit a fully completed "Prereg Challenge Registration" form for review by the COS. The submissions were reviewed for completeness of answering the questions and not for the substance or quality of the research. Prizes were earned after authors published their completed studies in 1 of the participating journals.

OSF now has templates for other preregistration formats including the Replication Recipe [24], the Preregistration in Social Psychology [23], the Registered Report Protocol [25,26], and the AsPredicted format (following the 8 question form provided at the AsPredicted.org

website), and many others are in development by communities of researchers for specific topical areas and methodologies such as neuroimaging and qualitative research [27–29]. Preregistration formats differ greatly in the extent to which they take the author by the hand in writing a preregistration that is sufficiently specific, precise, and exhaustive. Given the rapidly growing interest in preregistration across disciplines, it is important to evaluate the extent to which preregistrations restrict opportunistic use of researcher degrees of freedom. We hypothesize that more structured formats providing guidance and support for researchers will be more effective at reducing degrees of freedom than more unstructured formats.

In this study, we examined whether preregistrations prepared in a more "structured" format (Prereg Challenge Registrations) restrict opportunistic use of researcher degrees of freedom more than preregistrations prepared in an "unstructured" format (Standard Pre-Data Collection) that maximizes flexibility for the researcher to define preregistration content that is most fitting for their research. Furthermore, we investigated which researcher degrees of freedom are more restricted than others by the preregistrations. We did not examine the Open-ended Registrations because we wanted to only include registrations of which the researchers explicitly indicated that they had not collected or looked at the data before composing the preregistration. We also asked the managers of preregistration platform aspredicted.org to collaborate and include their preregistrations in our study, but they indicated that the public preregistrations would not be released until December 2016. As this would be after our data collection period, we decided not to assess their preregistrations in our study.

We evaluated to what extent preregistration formats restricted opportunistic use of 29 researcher degrees of freedom [18], collectively providing a Transparency Score for preregistrations. Specifically, we evaluated random samples of OSF "Standard Pre-Data Collection Registrations," hereafter "Unstructured," and "Prereg Challenge Registrations," hereafter "Structured," to (1) to test our preregistered confirmatory hypothesis that registrations completed in a Structured format would receive higher Transparency Scores on average than registrations completed in an Unstructured format; (2) to assess differences by format on each of the 29 researcher degrees of freedom; and (3) to use these findings to create preregistration guidelines that will restrict researcher degrees of freedom as effectively as possible. Note that these Transparency Scores were called "Restriction Scores" in the preregistration as it concerns descriptions that restrict opportunities for researcher degrees of freedom, but as these descriptions entail transparency about the research process, we use "Transparency Scores." We thank a reviewer for this suggestion.

The complete preregistration of our study can be found at https://osf.io/k94ve/. All deviations from the preregistration are presented at the end of the Methods section. To emulate our intended best practice on the 29 researcher degrees of freedom, we wrote our own preregistration according to these standards and continued revision until it received full marks from 1 of the members of the coding team (EC) who was not involved in the creation of the scoring protocol.

## Methods

### Researcher degrees of freedom assessed

To evaluate the extent to which Structured and Unstructured preregistration formats restricted opportunistic use of researcher degrees of freedom, we constructed a coding protocol based on 29 of the 34 degrees of freedom from Wicherts and colleagues ([18]; see Table 1). The items are categorized into 5 phases of the research process: formulating the hypotheses, designing the study, collecting the data, analyzing the data, and reporting. We excluded 5 items from the reporting phase of research (failing to assure reproducibility, failing to enable replication,

**Table 1. Degrees of freedom in formulating the hypotheses, designing the study, collecting the data, analyzing the data, and reporting of psychological studies.**

| Code | Related | Type of Researcher Degrees of Freedom | Label |
|------|---------|----------------------------------------|-------|
| **Hypothesizing** | | | |
| T1 | R6 | Conducting explorative research without any hypothesis | Hypothesis |
| T2 | | Studying a vague hypothesis that fails to specify the direction of the effect | Direction hypothesis |
| **Design** | | | |
| D1 | A8 | Creating multiple manipulated IVs and conditions | Multiple manipulated IVs |
| D2 | A10 | Measuring additional variables that can later be selected as covariates, IVs, mediators, or moderators | Additional IVs |
| D3 | A5 | Measuring the same DV in several alternative ways | Multiple measures DV |
| D4 | A7 | Measuring additional constructs that could potentially act as primary outcomes | Additional constructs |
| D5 | A12 | Measuring additional variables that enable later exclusion of participants from the analyses (e.g., awareness or manipulation checks) | Additional IVs exclusion |
| D6 | | Failing to conduct a well-founded power analysis | Power analysis |
| D7 | C4 | Failing to specify the sampling plan and allowing for running (multiple) small studies | Sampling plan |
| **Data Collection** | | | |
| C1 | | Failing to randomly assign participants to conditions | Random assignment |
| C2 | | Insufficient blinding of participants and/or experimenters | Blinding |
| C3 | | Correcting, coding, or discarding data during data collection in a non-blinded manner | Data handling/collection |
| C4 | D7 | Determining the data collection stopping rule on the basis of desired results or intermediate significance testing | Stopping rule |
| **Data Analysis** | | | |
| A1 | | Choosing between different options of dealing with incomplete or missing data on ad hoc grounds | Missing data |
| A2 | | Specifying preprocessing of data (e.g., cleaning, normalization, smoothing, and motion correction) in an ad hoc manner | Data preprocessing |
| A3 | | Deciding how to deal with violations of statistical assumptions in an ad hoc manner | Assumptions |
| A4 | | Deciding on how to deal with outliers in an ad hoc manner | Outliers |
| A5 | D3 | Selecting the DV out of several alternative measures of the same construct | Select DV measure |
| A6 | | Trying out different ways to score the chosen primary DV | DV scoring |
| A7 | D4 | Selecting another construct as the primary outcome | Select primary outcome |
| A8 | D1 | Selecting IVs out of a set of manipulated IVs | Select IV |
| A9 | D1 | Operationalizing manipulated IVs in different ways (e.g., by discarding or combining levels of factors) | Operationalizing manipulated IVs |
| A10 | D2 | Choosing to include different measured variables as covariates, IVs, mediators, or moderators | Include additional IVs |
| A11 | | Operationalizing non-manipulated IVs in different ways | Operationalizing non-manipulated IVs |
| A12 | D5 | Using alternative inclusion and exclusion criteria for selecting participants in analyses | In/exclusion criteria |
| A13 | | Choosing between different statistical models | Statistical model |
| A14 | | Choosing the estimation method, software package, and computation of SEs | Method and package |
| A15 | | Choosing inference criteria (e.g., Bayes factors, alpha level, sidedness of the test, corrections for multiple testing) | Inference criteria |
| **Reporting** | | | |
| R6 | T1 | Presenting exploratory analyses as confirmatory (HARKing) | HARKing |

Note: This table provides the codes used in the original list [18], indicates to which other degrees of freedom each degree of freedom is related, the description of the degree of freedom (identical to the original list), and short labels describing the degrees of freedom that we use later when describing our results.

DV, dependent variable; HARKing, hypothesizing after the results are known; IV, independent variable; SE, standard error.

failing to mention, misrepresenting or misidentifying the study preregistration, failing to report so-called "failed studies," and misreporting results and *p*-values) because they could not be assessed based on the preregistration.

## The scoring protocol

We created a protocol assessing to what extent a random selection of registrations from Structured and Unstructured formats restricted opportunistic use of the 29 degrees of freedom. We

---

## Box 2. Example coding protocol

Degree of freedom A4: "Deciding on how to deal with outliers in an ad hoc manner."

Does the preregistration indicate how to detect outliers and how they should be dealt with?

- **NO** not described at all → **A4 = 0**

- **PARTIAL** described but not reproducible on at least 1 of the following 2 aspects: what objectively defines an outlier (e.g., particular Z value, values for median absolute deviation (MAD) statistic, interquartile range (IQR), Mahalanobis distance) and how they are dealt with (e.g., exclusion, method of Winsorization, type of nonparametric test, type of robust method, and bootstrapping) → **A4 = 1**

- **YES** reproducible on both aspects (objective definition of outliers and method of dealing with outliers) → **A4 = 2**

- **YES** like previous AND explicitly excluding other methods of dealing with outliers ("we will only use") → **A4 = 3**

---

assigned scores from 0 to 3 to each degree of freedom: (0) not restricted at all; (1) restricted to some degree; (2) completely restricted (i.e., it was "specific" and "precise"); and (3) completely restricted and exhaustive (i.e., the preregistration included an explicit statement that no deviation from the way it was registered would occur). Box 2 provides an example coding protocol. There were some scoring dependencies among 13 of the degrees of freedom (see the protocol) such that some scores are correlated. Also, for 4 degrees of freedom, only scores of 0 or 3 were possible.

## Sample

At the start of our study (on August 17, 2016), 5,829 publicly available preregistrations were listed on the preregistrations search page on OSF. Of these, there were 122 public registrations in the Structured format and many in the Unstructured format. Following our preregistration, we randomly selected 53 Structured and 53 Unstructured registrations. This sample size was based on our preregistered power analysis to test the differences in median scores between the 2 types of preregistrations with a Wilcoxon–Mann–Whitney $U$ test (1-tailed), which was conducted in G*Power 3.1 [67] and yielded a total required sample size of 106 (53 per group) for a power of 0.80 to detect a medium effect size of Cohen's $d = 0.50$. We had no previous literature to base the estimated effect size on. Instead, we considered a medium effect size to be an indication of a practically relevant difference between the 2 types of preregistrations. Further, our protocol pretesting indicated that the average coding time per article was 2 hours. The practical constraints of time and resources held us to the sample size yielded by the power analysis.

## Procedure

**Selection of preregistrations.**   We selected the samples of preregistrations following our own preregistered plan with just 1 minor deviation from the plan (see the "Deviations from our preregistered protocol" section below).

---

## Coding procedure

Each preregistration was coded independently by 2 of the total 5 coders (CV, MB, MvA, HHO, and EC), according to a scheme generated by an R script (R version 3.2.4). The coders first entered the number of hypotheses they encountered in the preregistration into their coding sheet and then followed the scoring protocol. When coders finished, their scores were compared with an R script (R version 3.2.4). We then computed agreement percentage of scores and hypotheses counted with a third R script (R version 3.2.4).

Across all data, the same score had been given in 74.84% of the cases. For Unstructured formats, the same score had been given in 77.75% of the cases and for Structured formats in 71.88%. Coders agreed on the number of hypotheses in only 14.29% of the scores. Across Unstructured formats, this agreement percentage was 15.09%, and across Structured formats, this was 13.46%.

For coding discrepancies, the 2 coders discussed until they agreed on a final score. This discussion was sufficient to resolve all discrepancies, thus no third coder was needed to solve a disagreement. We did not attempt to resolve discrepancies about the number of hypotheses as this was not part of our analyses but merely served as an indication of clarity and specificity of the preregistrations. The coders were not blinded for registration type as they visibly differ in structure.

## Variables of interest

Following our preregistration, we computed a score indicating to what extent a preregistration restricted opportunistic use of degrees of freedom. This Transparency Score was computed as the unweighted arithmetic mean of the scores (0 to 3) of all 29 researcher degrees of freedom in our protocol. Some degrees of freedom carried more weight in the Transparency Score than others because of dependencies between them (see Table 1). The "means per degree of freedom" in Table 2 were calculated as the unweighted arithmetic mean of the scores (0 to 3) across each set of 53 and 52 preregistrations.

A reviewer of a prior version of this manuscript argued that 3 items (D6, failing to conduct a power analysis; C1, failing to randomly assign participants to conditions; and C2, insufficient blinding) may not be considered researcher degrees of freedom but are rather choices that affect the quality of a study (see Bakker and colleagues [40] for an investigation of the relationship between preregistration and statistical power in the current sample). Therefore, for exploratory purposes, we also calculated an aggregate score based on all items except those 3. Furthermore, we calculated Transparency Scores for each subcategory (e.g., design and data collection) separately. Note that Transparency Scores based on these subsets are not preregistered, and all analyses that include these scores should be considered exploratory.

## Statistical analyses

To test our primary hypothesis, we compared the median Transparency Score of the Structure format to the median Transparency Score of the Unstructured format with a 1-tailed 2-group Wilcoxon–Mann–Whitney $U$ test. We chose this test because we expected the Transparency Score to be non-normally distributed, and this nonparametric test is robust against non-normality while still being relatively powerful [41]. We then conducted preregistered follow-up analyses to investigate on which degrees of freedom the 2 preregistration types differed. We conducted 29 preregistered 2-tailed Wilcoxon–Mann–Whitney $U$ tests to compare the median scores of the 2 formats per degree of freedom. In all analyses, we maintained an inference criterion of alpha = 0.05. In addition, we conducted registered follow-up analyses with 2-tailed

**Table 2. Means and distributions of scores per degree of freedom for registrations from Unstructured and Structured formats and differences in median scores between formats.**

| DF | | Unstructured Format | | | | | | Structured Format | | | | | | Test | Differences in Median | |
|----|----|----|----|----|----|----|----|----|----|----|----|----|----|----|----|----|
| | | Mean (SD) | 0 | 1 | 2 | 3 | NA | Mean (SD) | 0 | 1 | 2 | 3 | NA | | Holm p | Cliff's D |
| **Hypothesizing** | | | | | | | | | | | | | | | | |
| T1 | Hypothesis | 1.98 (0.31) | 1.9 | - | 96.2 | 1.9 | - | 2.02 (0.14) | 0.0 | - | 98.1 | 1.9 | - | W = 1,404, p = 0.571 | 1.000 | 0.02 |
| T2 | Direction hypothesis | 1.60 (0.84) | 20.8 | - | 77.4 | 1.9 | - | 1.54 (1.20) | 34.6 | - | 42.3 | 23.1 | - | W = 1,422, p = 0.749 | 1.000 | 0.03 |
| **Design** | | | | | | | | | | | | | | | | |
| D1 | Multiple manipulated IVs | 0.38 (1.02) | 64.2 | - | 0.0 | 9.4 | 26.4 | 1.03 (1.42) | 46.2 | - | 1.9 | 23.1 | 28.8 | W = 880, p = 0.026 | 0.443 | 0.22 |
| D2 | Additional IVs | 0.00 (0.00) | 100 | - | - | 0.0 | - | 0.12 (0.58) | 96.2 | - | - | 3.8 | - | W = 1,431, p = 0.155 | 1.000 | 0.04 |
| D3 | Multiple measures DV | 1.25 (0.98) | 37.7 | - | 62.3 | 0.0 | - | 1.62 (0.80) | 19.2 | - | 80.8 | 0.0 | - | W = 1,633, p = 0.037 | 0.540 | 0.19 |
| D4 | Additional constructs | 0.00 (0.00) | 100 | - | - | 0.0 | - | 0.00 (0.00) | 100 | - | - | 0.0 | - | NA | NA | 0.00 |
| D5 | Additional IVs exclusion | 0.87 (0.92) | 45.3 | 26.4 | 24.5 | 3.8 | - | 1.23 (0.70) | 13.5 | 51.9 | 32.7 | 1.9 | - | W = 1,729.5, p = 0.017 | 0.327 | 0.26 |
| D6 | Power analysis | 0.72 (0.91) | 58.5 | 11.3 | 30.2 | 0.0 | - | 0.96 (0.99) | 50.0 | 3.8 | 46.2 | 0.0 | - | W = 1,551, p = 0.212 | 1.000 | 0.13 |
| D7 | Sampling plan | 0.47 (0.58) | 56.6 | 39.6 | 3.8 | 0.0 | - | 0.71 (0.58) | 34.6 | 57.7 | 5.8 | 0.0 | 1.9 | W = 1,641, p = 0.034 | 0.540 | 0.21 |
| **Data Collection** | | | | | | | | | | | | | | | | |
| C1 | Random assignment | 0.27 (0.67) | 66.0 | 1.9 | 9.4 | 0.0 | 22.6 | 0.86 (0.92) | 34.6 | 11.5 | 25.0 | 0.0 | 28.8 | W = 1,028.5, p = 0.001 | 0.023 | 0.36 |
| C2 | Blinding | 1.00 (1.00) | 3.8 | 1.9 | 3.8 | 0.0 | 90.6 | 0.02 (0.14) | 92.3 | 1.9 | 0.0 | 0.0 | 5.8 | W = 50.5, p < 0.001 | <0.001 | −0.59 |
| C3 | Data handing/collection | 0.04 (0.19) | 96.2 | 3.8 | 0.0 | 0.0 | - | 0.04 (0.19) | 96.2 | 3.8 | 0.0 | 0.0 | - | W = 1,379, p = 0.992 | 1.000 | 0.00 |
| C4 | Stopping rule | 0.47 (0.58) | 56.6 | 39.6 | 3.8 | 0.0 | - | 0.71 (0.58) | 34.6 | 57.7 | 5.8 | 0.0 | 1.9 | W = 1,641, p = 0.034 | 0.540 | 0.21 |
| **Data Analysis** | | | | | | | | | | | | | | | | |
| A1 | Missing data | 0.19 (0.39) | 81.1 | 18.9 | 0.0 | 0.0 | - | 0.76 (0.55) | 28.8 | 63.5 | 5.8 | 0.0 | 1.9 | W = 2,065.5, p < 0.001 | <0.001 | 0.53 |
| A2 | Data preprocessing | 0.50 (0.84) | 9.4 | - | 1.9 | 0.0 | 88.7 | 0.50 (0.93) | 11.5 | - | 3.8 | 0.0 | 84.6 | W = 23, p = 0.935 | 1.000 | −0.04 |
| A3 | Assumptions | 0.04 (0.19) | 96.2 | 3.8 | 0.0 | 0.0 | - | 0.18 (0.48) | 84.6 | 9.6 | 3.8 | 0.0 | 1.9 | W = 1,488, p = 0.070 | 0.835 | 0.10 |
| A4 | Outliers | 0.25 (0.62) | 84.9 | 5.7 | 9.4 | 0.0 | - | 0.69 (0.92) | 57.7 | 19.2 | 19.2 | 3.8 | - | W = 1,751, p = 0.003 | 0.056 | 0.27 |
| A5 | Select DV measure | 1.25 (0.98) | 37.7 | - | 62.3 | 0.0 | - | 1.62 (0.80) | 19.2 | - | 80.8 | 0.0 | - | W = 1,633, p = 0.037 | 0.540 | 0.19 |
| A6 | DV scoring | 0.55 (0.70) | 56.6 | 32.1 | 11.3 | 0.0 | - | 0.65 (0.65) | 44.2 | 46.2 | 9.6 | 0.0 | - | W = 1,519, p = 0.317 | 1.000 | 0.10 |
| A7 | Select primary outcome | 0.00 (0.00) | 100 | - | - | 0.0 | - | 0.00 (0.00) | 100 | - | - | 0.0 | - | NA | NA | 0.00 |
| A8 | Select IV | 0.59 (1.19) | 58.5 | - | 1.9 | 13.2 | 26.4 | 1.14 (1.48) | 44.2 | - | 0.0 | 26.9 | 28.8 | W = 853.5, p = 0.083 | 0.910 | 0.18 |
| A9 | Operationalize manipulated IVs | 1.05 (1.26) | 41.5 | - | 18.9 | 13.2 | 26.4 | 1.92 (1.19) | 17.3 | - | 25.0 | 28.8 | 28.8 | W = 982.5, p = 0.004 | 0.078 | 0.36 |
| A10 | Include additional IVs | 0.00 (0.00) | 100 | - | - | 0.0 | - | 0.12 (0.58) | 96.2 | - | - | 3.8 | - | W = 1,431, p = 0.155 | 1.000 | 0.04 |
| A11 | Operationalize non-manipulated IVs | 0.43 (0.66) | 28.3 | 11.3 | 3.8 | 0.0 | 56.6 | 0.63 (0.67) | 26.9 | 25.0 | 5.8 | 0.0 | 42.3 | W = 405, p = 0.229 | 1.000 | 0.17 |
| A12 | In/exclusion criteria | 0.87 (0.92) | 45.3 | 26.4 | 24.5 | 3.8 | - | 1.21 (0.72) | 15.4 | 50.0 | 32.7 | 1.9 | - | W = 1,710.5, p = 0.024 | 0.438 | 0.24 |
| A13 | Statistical model | 0.85 (0.77) | 37.7 | 39.6 | 22.6 | 0.0 | - | 1.31 (0.51) | 1.9 | 65.4 | 32.7 | 0.0 | - | W = 1,846, p = 0.001 | 0.023 | 0.34 |
| A14 | Method and package | 0.08 (0.38) | 96.2 | 0.0 | 3.8 | 0.0 | - | 0.13 (0.44) | 90.4 | 5.8 | 3.8 | 0.0 | - | W = 1,455.5, p = 0.254 | 1.000 | 0.06 |
| A15 | Inference criteria | 0.17 (0.43) | 84.9 | 13.2 | 1.9 | 0.0 | - | 1.08 (0.33) | 1.9 | 88.5 | 9.6 | 0.0 | - | W = 2,516, p < 0.001 | <0.0001 | 0.83 |
| **Reporting** | | | | | | | | | | | | | | | | |
| R6 | HARKing | 0.00 (0.00) | 100 | - | 0 | 0.0 | - | 0.00 (0.00) | 100 | - | 0.0 | 0.0 | - | NA | NA | 0.00 |

Note: The mean scores per degree of freedom can range from 0 to 3. Distribution of scores are given in percentages. Not all percentages add up to exactly 100% due to rounding to 1 decimal of each individual percentage. A "-" sign indicates that this score was not possible for this degree of freedom (see Methods section).

Cliff's D, Cliff's Delta; DF, degree of freedom; DV, dependent variable; HARKing, hypothesizing after the results are known; Holm p, Holm p-value; IV, independent variable; NA, not applicable; SD, standard deviation.

paired-sample sign tests (without ties) to examine which degrees of freedom are the least and most restricted across both formats.

For exploratory analyses, we performed a nonparametric bootstrap procedure to compare the median Transparency Scores. Furthermore, the Transparency Scores on the subsets of items created for exploratory purposes are compared with the 2-tailed 2-group Wilcoxon–Mann–Whitney U test consistent with the other analyses. We explored the association between the Transparency Scores of the different categories with Spearman rank order correlations.

We employed Cliff's Delta [42,43] to assess effect size for comparing central tendency of ordinal variables. Cliff's Delta does not make any assumptions on the distributions of the 2 variables and is easily interpretable. Values under 0.147 are considered "negligible," values between 0.147 and 0.330 are considered "small," values between 0.330 and 0.474 are considered "medium," and values larger than 0.474 are considered "large" [44]; see https://cran.r-project.org/web/packages/effsize/effsize.pdf. Although we did not preregister a correction for multiple comparisons, we present the Holm corrected *p*-values, taking into account the large number of tests we conducted.

For our analyses, we used an R script (R version 3.2.4), which was an elaborated version of our preregistered analysis script. We did not exclude any data except for 1 preregistration before coding it (see sample description). The data file was checked for missing values, and coders who left values missing then coded the variables they missed. Values coded as 99 indicated that the specific variable was not applicable for that registration. Following our preregistration, we employed 2-way imputation (based on corresponding row and column means) to handle missing values. And, for our planned follow-up analyses, we employed pairwise deletion of missing values at the degree of freedom level. The tables in this manuscript were created using an R Markdown script. The degree of freedom labels were added manually, and some column names were adapted. The R Markdown script can be used directly when loading the workspace of the analysis script.

### Deviations from our preregistered protocol

We started the coding with 53 preregistrations from each format, but then discovered 1 from the Structured format (#54) had been withdrawn by its authors. We therefore excluded this preregistration from our data file. Our final sample thus consisted of 53 in the Unstructured format condition and 52 in the Structured format condition.

Following our initial selection strategy drawing 250 preregistrations and then coding for eligibility, we selected only 31 registrations for the Unstructured format condition, fewer than our target sample. To achieve our target sample, we randomly selected a second sample of 250 preregistrations with the same code and applied the same procedures, which resulted in 29 preregistrations. Checking for duplicates, we found 6 preregistrations for which the hyperlink referred to the same project as another preregistration's hyperlink. Ordering all 54 registrations from first to last coded, we selected the first 53 out of 54 remaining to include in our final sample.

## Results

### Main analysis: Overall difference between registrations created with Unstructured versus Structured formats

Our first research question was whether preregistrations that were written following a more detailed and specific form (Structured format) restricted use of researcher degrees of freedom more than preregistrations that were written following an open and flexible form (Unstructured format). In line with our confirmatory hypothesis, registrations from Structured formats (Mdn = 0.81) received higher median Transparency Scores than those from Unstructured formats (Mdn = 0.57), U = 2,053, *p* < 0.001. The difference was large (Cliff's Delta = 0.49). The highest Transparency Score of a preregistration received in the Unstructured format was 1.05, whereas the highest score received in the Structured group was 1.47 (Range = 0 to 3).

## Follow-up analyses: Differences between registrations created with Unstructured and Structured formats for each degree of freedom

In Table 2 and S1 and S2 Figs, we report for each type of preregistration the mean and distribution of scores for each degree of freedom and the Wilcoxon–Mann–Whitney *U* test and Cliff's Delta effect size. We observed higher scores in Structured than Unstructured formats for 22 of 29 degrees of freedom and significantly so for 14 of 29 ($\alpha = 0.05$) and for 5 of 29 after applying the Holm correction (note that this correction was not preregistered). For just 1 degree of freedom (blinding) was the score significantly higher for Unstructured than Structured formats. Scores of 3 were rare indicating that use of researcher degrees of freedom was not fully restricted (specific, precise, and exhaustive) in either format.

**Degrees of freedom related to the hypothesis (and reporting).** All preregistrations scored 0 on restricting hypothesizing after the results are known (HARKing; R6), restricting measuring additional constructs (D4), and restricting the selection of another primary outcome (A7). Scores for these could only be 0 or 3, and none of the preregistrations explicitly specified that the confirmatory analysis section of the paper would not include another dependent variable than the ones specified in the preregistration. An obvious possible explanation is that researchers may assume that not mentioning other dependent variables means that no other dependent variables exist.

**Degrees of freedom related to design.** Registrations from Structured formats performed better than those from Unstructured formats on degrees of freedom that pertained to the operationalization of the variables in the study. For example, Structured formats outperformed Unstructured formats on preventing the inclusion of multiple manipulated independent variables (D1; D = 0.22), multiple measures of the dependent variables (D3 and A5; D = 0.19), and additional independent variables that can be used for inclusion and exclusion (D5 and A12; D = 0.24). Transparent reporting of these degrees of freedom can prevent common questionable research practices like failing to report all dependent variables, failing to report all of a study's conditions, and informed exclusion of data (self-admittances rates of 63.4%, 27.7%, and 38.2%, respectively; [17]). Likewise, clarity about the sampling plan (D7; D = 0.21) and data collection stopping rule (C4; D = 0.21) were better in the Structured format. The Structured format leaves room for authors to state that they will recruit a sample of "at least" a certain size. Therefore, continued sampling after intermediate testing is not precluded, which might explain that the difference here was rather small.

Most of the degrees of freedom related to design were relatively well restricted compared to others, although restricting the inclusion of multiple manipulated independent variable (D1) and restricting the inclusion of additional variables that could be used for covariates or moderators (D2 and A10) were not in either format. The Structured format asks authors to list all variables in the study, but it does not ask for what purpose.

**Degrees of freedom related to data collection.** Transparency about random assignment (C1) was rather poor in both formats, but better in Structured (D = 0.36), in which the format prompts authors to at least mention randomization. Transparency of how data are treated during data collection was hardly addressed in either format, and no difference was observed (C3; D = 0.00).

Curiously, reporting about blinding participants or experimenters (C2) was better in the Unstructured than Structured format, and the difference was large (D = −0.59). However, this may have occurred because of the coding strategy. In most Unstructured registrations (90.6%), this degree of freedom was coded as "Not Applicable," whereas in most Structured registrations (92.3%), it was coded as 0. The protocol first asked whether blinding of participants and/ or experimenters was mentioned. In Unstructured registrations, the word "blinding" hardly

ever appeared, resulting in most being coded as "not applicable." If blinding was mentioned, the coder checked whether the preregistration described procedures to blind participants or experimenters. This was often coded as 0 because authors responded that blinding was not applicable or that no blinding occurred. We decided to adhere to the preregistered protocol; however, we believe the resulting scores are unrepresentative of the Structured formats performance on this item.

**Degrees of freedom related to data analysis.** Transparency about testing and handling the assumptions of the statistical model (A3) and the software and packages to use for analysis (A14) was poor in both formats, and there was no significant difference ($D = 0.10$ and $D = 0.06$, respectively). The Structured format was rather specific about aspects to be considered in the description of the statistical model, but it does not explicitly solicit information about violations of statistical assumptions, estimation method, or software to be used. Neither format did very well in eliciting complete responses for handling missing data (A1), but the Structured format did much better than the Unstructured format ($D = 0.53$). Both formats did better in reporting how outliers would be managed (A4), with the Structured format doing better than the Unstructured format ($D = 0.27$).

The Structured format also did better in eliciting clarity about operationalizing of independent variables (A9) compared with the Unstructured format ($D = 0.36$). However, the mean scores on this degree of freedom were heavily influenced by the relatively high number of preregistrations that received a score of 3, particularly for the Structured format. When the analysis concerned a *t* test (or a nonparametric equivalent), a common occurrence, we considered this degree of freedom excluded by definition and therefore assigned a score of 3. Also, the Structured format encouraged and elicited sharing of analysis scripts (leading to a score of at least 2), whereas this never occurred in the Unstructured format. The choice of statistical model (A13) was relatively well reported in both formats, but better in the Structured format ($D = 0.34$). Finally, the criteria for drawing inference (A15) was much better reported in the Structured than Unstructured format ($D = 0.83$). Degrees of freedom associated with a specific prompt in the Structured format tended to show the strongest performance advanced for that format.

## Exploratory analyses

We explored which of the degrees of freedom indicators tended to perform better and worse than the others across registration formats. This can help inform where preregistration is doing relatively well in transparent reporting and where there is most opportunity to improve. Exploratory 2-tailed paired-sample sign tests revealed many overlapping subsets that were statistically indistinguishable. However, transparent reporting of the hypotheses (T1) was significantly better than all other degrees of freedom. Both formats completely restricted (a Transparency Score of 2) this degree of freedom (Unstructured: mean = 1.98; Structured: mean = 2.02). This can be explained by our selection of the preregistrations, which should contain at least 1 statistically testable hypothesis. The next 4—clarifying hypothesis direction (T2), restricting the use of multiple dependent variables (D3 and A5), and restricting operationalizing manipulated independent variables in different ways (A9)—were similar to each other and significantly better than all the rest. Eight degrees of freedom showed significantly worse performance than the rest: Include (D2) and select additional independent variables (A10), include (D4) and select another primary outcome (A7), how data were handled during data collection (C3), how assumptions about analysis strategy were tested and addressed (A3), the analysis software and packages used (A14), and addressing hypothesizing after the results were known (R1). All of these had scores near 0 suggesting that they were hardly addressed at all.

We computed Transparency Scores on the subset that follow more strictly the definition of researcher degrees of freedom (i.e., excluding items D6, C1, and C2). As in the full analysis, the Structured format (Mdn = 0.83) performed better than the Unstructured format (Mdn = 0.54), U = 2,090, $p < 0.001$. The difference was large (Cliff's Delta = 0.52) and similar in magnitude when all measures were included (Cliff's Delta = 0.49). These follow-up and exploratory analyses are tested 2-tailed. An additional exploratory bootstrap procedure compared the medians of original Transparency Scores is consistent with the preregistered result ($p = 0.001$, 2-tailed).

Transparency Scores were calculated for each subcategory of the coding measure. Two-tailed Wilcoxon–Mann–Whitney $U$ tests showed significant large differences in Transparency Scores for the subcategory Design (Unstructured mdn = 0.57; Structured mdn = 0.86; U = 1,930, $p < 0.001$; Cliff's Delta = 0.40) and Data Analysis (Unstructured mdn = 0.82; Structured mdn = 0.49; U = 2,148, $p < 0.001$; Cliff's Delta = 0.56). The Design Transparency Score was correlated with the Data Collection Transparency Score ($r_s = 0.478$, $p < 0.001$) and with the Data Analysis Transparency Score ($r_s = 0.794$, $p < 0.001$). Data Collection Transparency Scores were correlated with Data Analysis Transparency Scores ($r_s = 0.354$, $p < 0.001$).

## General discussion

We observed that preregistrations written with a more detailed and guided workflow (Structured format) restricted use of researcher degrees of freedom better than preregistrations written with a more flexible and open-ended workflow (Unstructured format). This was directionally observed for 22 of 29 degrees of freedom, but only 14 of 29 were significantly so ($p < 0.05$), and only 5 were statistically significant after Holm correction (exploratory analysis). The median effect size by degree of freedom was 0.17 and only 2 showed large effects in the expected direction (Cliff's Delta > 0.474): missing data (A1) and inference criteria (A15). The Structured format appeared to outperform the Unstructured format more when its prompts specifically articulated instructions to reduce degrees of freedom. We conclude that preregistrations are likely to be more effective when using protocols with specific, comprehensive instructions about what needs to be reported.

The Structured format outperformed the Unstructured format, but neither performed impressively. The median Transparency Score for Unstructured was 0.57 and for Structured was 0.81, on a 0 (worst) to 3 (best) scale. In many cases, the Structured format hardly performed better than the Unstructured format, and 8 degrees of freedom indicators scored close to 0 on average. Most often, low scores occurred when the degree of freedom was not explicitly addressed in the Structured workflow reporting instructions.

On the other hand, the modest overall performance is partly attributable to the strictness of our protocol. The highest rating of "3" required an "exhaustive" description explicitly excluding the possibility of deviations from the preregistration. It is reasonable to argue that preregistration is explicitly restricting deviations by definition, obviating the need for researchers to make it explicit. However, we do not know this for sure. For example, if a researcher does not state anything about the handling of outliers, it could mean they will not remove any, or it could mean that they will make decisions about outliers after the fact. Studies on registrations of clinical trials show that outcome switching and HARKing are common [45–55], perhaps because of a failure to commit to not doing these behaviors. Explicit commitments to exhaustiveness—what will not be done—may reduce this tendency. Straightforward adaptations of preregistration workflows may be helpful. For example, preregistration workflows could add checkboxes to confirm intentions not to deviate from the preregistered plan and commitments to explicitly report any unavoidable deviations.

## Limitations

We interpret our findings as indicating that more structured preregistration protocols will improve transparency, restrict researcher degrees of freedom, and, ultimately, improve the credibility of preregistered findings. An obvious limitation is that researchers were not randomly assigned to complete Structured or Unstructured formats. Our causal conclusion relies on plausibility of the causal scenarios for the association. It is possible that researchers who desire to retain flexibility and preregister studies for performative rather than substantive purposes would select unstructured formats, and researchers who desire to meet the promise of preregistration would select structured formats. We find this to be implausible. We perceive that researchers uncommitted to the purpose of preregistration would just decide to not preregister. Also, having observed and coded the preregistrations, we perceive uses of the unstructured format to be mostly serious and genuine, just incomplete. Nevertheless, we cannot rule out causal scenarios different than our proposal that providing structured format guides researchers to be more explicit and transparent with their research plans.

Another limitation is that coders could not be blinded for registration type. The difference in structure is obvious and part of the phenomenon being studied. As a consequence, our coding could have been influenced by knowledge of condition. We attempted to minimize this possibility with multiple coders and by making the protocol as objective as possible. In the event that these efforts were insufficient, by working transparently and making all data and material available, we make it possible to independently verify our coding and results.

Another limitation is that the Structured format registrations we coded had been reviewed for completeness by COS staff when entered the Preregistration Challenge. That review was not substantive about the content of the preregistration, but it did involve assessing whether the study met criteria for entry into the Challenge (i.e., that all required fields were completed, that the study was inferential, and that the plan was reasonably clear; see https://osf.io/h4ga8/ for the reviewer instructions). It is possible that the review elicited feedback that improve explicitness of the preregistrations or that researchers would have left more questions unanswered had there not been a review process. It is notable, however, that even the Structured format registrations have substantial room to improve, minimally suggesting that the format and this mild review are insufficient. We do expect that peer review of preregistration of protocols can contribute positively toward improving their transparency, specificity, and exhaustiveness. A productive step for follow-up research would be to examine the independent and interactive roles of improved structure of the registration form and workflow and a relatively intensive peer review process, such as through the Registered Reports publishing format [56]. We hypothesize that both factors would make positive contributions to the rigor of preregistrations, and it would be productive to understand the value added by peer review in comparison to the resources required to provide that service.

Another limitation is that most preregistrations (88.6%) were from psychology. This partly reflects the fact that preregistrations popularity has grown most quickly in this field. The researcher degrees of freedom coding format was developed in the context of social and behavioral research providing good matching for the purposes of this research [18]. To evaluate the generalizability of our findings to other fields, it will be worth reviewing and updating the coding format for unique methodological issues in other disciplines. New registration formats have emerged for other research activities such as animal studies (https://www.animalstudyregistry.org; https://www.preclinicaltrials.eu), qualitative research [30], neuroimaging [29], and cognitive modeling [28]. These offer productive opportunities for extending this research to new domains and preregistration formats (see https://osf.io/zab38/ for a curated list).

A final limitation is the comprehensiveness and reliability of our protocol. We coded the preregistration itself and not the correspondence between the preregistration and the final report, another area in which researchers could intentionally or unintentionally exploit researcher degrees of freedom [57,58]. Also, coders agreed on 74% of the scores suggesting some coding challenges. For some preregistrations, the protocol was difficult to apply (e.g., studies that used secondary data or Bayesian statistics). Also, some preregistrations were so ambiguous that it was hard to make sense of the planned research. For example, it was frequently difficult to understand what the main hypotheses were (e.g., multiple dependent variables are stated in 1 sentence, which might indicate 1 multivariate hypothesis, or multiple hypotheses that are tested with separate tests), as demonstrated by the percentage of agreement on the number of hypotheses in a preregistration being extremely low (around 14%). This might indicate that raters need more training. However, other studies also had difficulties with coding hypotheses. For example, Hartgerink and colleagues [59] found that only in 15 of 178 gender effects reported in published studies clearly stated whether the effects were as expected or hypothesized. Motyl and colleagues [60] had difficulty selecting statistical results as focal results [61], and researchers experienced some difficulties in specifying the main hypothesis in the 100 primary studies included in the Reproducibility Project: Psychology [62]. We interpret this as evidence that clearly stating testable hypotheses and expectations is surprisingly challenging and an area for substantial improvement [63].

## Next steps

Based on our findings, we can suggest improvements to the preregistration process to better restrict researcher degrees of freedom. First, preregistration formats need to help authors with instructions that are clear, precise, and exhaustive. Researchers can be prompted to explicitly state that the stated plans are comprehensive. Second, preregistrations could prompt researchers to explicitly articulate each hypothesis and how it will be tested. Drawing a link between hypothesis and analysis will clarify how the researcher will draw inferences from the results. Third, preregistration may benefit from formal peer review before collection of the data. The review of Structured protocols in this research did not include a substantive review of the design, hypotheses, and analysis plan. A more substantive review could have added benefits like identifying lack of clarity on degrees of freedom that authors miss, such as through the Registered Reports publishing format ([26,56, 64]; http://cos.io/rr/).

It is reasonable to wonder if the benefits of detailed preregistration exceed the costs of preparing the preregistration. We cannot answer this question with the present data. However, there are good reasons to believe that the specification of design and analysis plans in advance will increase the rigor of the research and quality and reproducibility of the reported confirmatory evidence [1,2]. We experienced some of the benefits ourselves in the design of the preregistration for this study. The collaborative discussion of the preregistration clarified and aligned research objectives among the team, identified resource limitations and shaped strategy for maximizing the value of the design against the resources available, and helped specify precisely how we would conduct the analysis to test our hypotheses. That specificity increased our confidence in our findings and informed how we reported the results. Moreover, we observed efficiency benefits later in the research process because we already knew how we would analyze the data and report the results, and we knew very clearly when we completed the confirmatory phase of the research and entered exploratory analysis. Deviations from our plan were transparent and knowing those deviations prompted productive reflection to calibrate their potential impact on our findings and interpretation. All told, we hypothesize that preregistration provides a substantial efficiency benefit for research [65].

Recently, participants who had completed a preregistration were asked about their experiences with preregistration in a survey [66]. According to the participants, "preregistration encourages a more thoughtful planning process which leads to better-executed studies" (p. 19; [66]), and they expected that it will increase the confidence in the study by journal editors. On the other hand, participants of this survey also noted some possible disadvantages, like the amount of time it takes to write a preregistration and that they are unsure what to do with (reasonable) changes from the preregistered plan, which shows the need for more guidance of researchers who want to preregister a study. It is also not known yet, whether preregistrations put a substantial extra burden on reviewers. It will be productive to investigate potential trade-offs of overall research efficiency, quality, and progress in the context of preregistration and the extent of their specificity and exhaustiveness.

## Conclusions

Preregistration does not imply that confirmatory analyses are good, and exploratory analyses are bad. The goal of preregistration is to be transparent and make clear which is which. When there is no clear theory, or when the data or the analyses are quite complex, it could be more efficient to begin with exploratory analysis despite the costs to diagnosticity of statistical inferences. Further, exploratory analysis is important and valuable for generating hypotheses from observed data. Indeed, many important discoveries emerged from exploratory analysis in an unanticipated pattern of results. This can lead to a new hypothesis or a new analysis method that is then tested in new data. Preregistration does not tie the researchers' hands; it makes the research process transparent toward more trustworthy and replicable results. Having specific, precise, and comprehensive preregistrations and efficient workflows that support producing them may facilitate better understanding and rigor of exploratory and confirmatory research and ultimately accelerate progress.

## Supporting information

**S1 Fig. Distributions of scores per degree of freedom for Unstructured (U) and Structured (S) registration formats (part 1).** The data underlying this Figure can be found at https://osf. io/fgc9k/.
(TIFF)

**S2 Fig. Distributions of scores per degree of freedom for Unstructured (U) and Structured (S) registration formats (part 2).** The data underlying this Figure can be found at https://osf. io/fgc9k/.
(TIFF)

## Author Contributions

**Conceptualization:** Marjan Bakker, Coosje L. S. Veldkamp, Marcel A. L. M. van Assen, Brian A. Nosek, Courtney K. Soderberg, Jelte M. Wicherts.

**Formal analysis:** Marjan Bakker, Coosje L. S. Veldkamp, Marcel A. L. M. van Assen.

**Investigation:** Marjan Bakker, Coosje L. S. Veldkamp, Marcel A. L. M. van Assen, Elise A. V. Crompvoets, How Hwee Ong, David Mellor.

**Methodology:** Marjan Bakker, Coosje L. S. Veldkamp, Marcel A. L. M. van Assen, Jelte M. Wicherts.

**Project administration:** Coosje L. S. Veldkamp.

**Resources:** Marjan Bakker, Coosje L. S. Veldkamp, Marcel A. L. M. van Assen, Jelte M. Wicherts.

**Writing – original draft:** Marjan Bakker, Coosje L. S. Veldkamp, Marcel A. L. M. van Assen, Brian A. Nosek, Jelte M. Wicherts.

**Writing – review & editing:** Marjan Bakker, Coosje L. S. Veldkamp, Marcel A. L. M. van Assen, Elise A. V. Crompvoets, How Hwee Ong, Brian A. Nosek, Courtney K. Soderberg, David Mellor, Jelte M. Wicherts.

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
