## [Editor Report · Decision Letter 0]

2 Mar 2020

Dear Dr Bakker, 

Thank you for submitting your manuscript entitled "Ensuring the quality and specificity of preregistrations" for consideration as a Meta-Research Article by PLOS Biology.

Your manuscript has now been evaluated by the PLOS Biology editorial staff as well as by an Academic Editor with relevant expertise. I apologize for the delay while we discussed the information provided. I am writing to now let you know that we would like to consider this manuscript further. While we will take the previous reviews from Advances in Methods and Practices in Psychological Science into consideration, unfortunately since two of the reviewers do not reveal their identity we are not able to fully utilize those reviews and will need to get 1-2 reviews of our own. If you are happy with this, please proceed with the steps below.

Before we can send your manuscript to reviewers, we need you to complete your submission by providing the metadata that is required for full assessment. To this end, please login to Editorial Manager where you will find the paper in the 'Submissions Needing Revisions' folder on your homepage. Please click 'Revise Submission' from the Action Links and complete all additional questions in the submission questionnaire.

Please re-submit your manuscript within two working days, i.e. by Mar 04 2020 11:59PM.

Kind regards,

Hashi Wijayatilake, PhD,

Managing Editor

PLOS Biology

---

## [Decision Letter · Decision Letter 1]

13 May 2020

Dear Dr Bakker,

Thank you very much for submitting your manuscript "Ensuring the quality and specificity of preregistrations" for consideration as a Meta-Research Article at PLOS Biology. Your manuscript has been evaluated by the PLOS Biology editors, an Academic Editor with relevant expertise, and by two PLOS Biology reviewers. Firstly, let me sincerely apologize for the unusual and lengthy delay getting you this decision. As I had explained in an earlier email, we had a reviewer signed up who, despite assurances, never delivered his/her comments. We then had to get new reviewers signed on at that late stage. We had two reviewers signed on with the aim of getting 1-2 additional reviewers, as mentioned, and luckily both have now delivered on schedule. The good news is that both are quite positive about the study and find the research question interesting. Considering these reviews, along with the reviews from Advances in Methods and Practices in Psychological Science, especially Reviewer 3 (who had signed his review), we will be happy to pursue the manuscript at PLOS Biology. However, the reviewers do raise important points that will need to be thoroughly addressed in a revision. Primarily, Reviewer 2 has concerns about the analysis and power calculations, which will need to be fully and satisfactorily addressed. Both reviewers also point out that, given the broad potential interest in this article across fields, it needs to be made much more understandable. The editorial team and Academic Editor all agree that a major rewrite is needed for this to be suitable for our broad readership. Overall, while we will not be able to accept the current version of the manuscript, we would welcome re-submission of a revised version that takes into account the reviewers' comments. 

Please note that we cannot make any decision about publication until we have seen the revised manuscript and your response to the reviewers' comments. Your revised manuscript is also likely to be sent for further evaluation by the reviewers.

We expect to receive your revised manuscript within 2 months. 

**IMPORTANT - SUBMITTING YOUR REVISION**

*Re-submission Checklist*

*Published Peer Review*

*PLOS Data Policy*

*Blot and Gel Data Policy*

Sincerely,

Hashi Wijayatilake, PhD, 

Managing Editor

PLOS Biology

REVIEWS:

Reviewer #1 (Matthew Page): 

This is a really interesting study addressing an important question: does a more detailed template for registering study plans lead to a more transparent plan? The methods are rigorous and documented in sufficient detail to allow others to replicate the study. The authors did a great job responding to the reviewer comments at the other journal, but I have a few other comments that I hope will improve the manuscript.

General comment: The term preregistration is used throughout. I am from medicine where we have used the terms prospective registration or retrospective registration for decades, so I find the "pre" added to this term confusing. The use of "pre" implies some act done before registering a study, but I'm not sure what that act is. A clarification about terminology would be great, particularly for health and medical researchers who would also benefit from reading this work.

Abstract: The authors claim that "Opportunistic use of 'researcher degrees of freedom' aimed at obtaining statistical significance increases the likelihood of obtaining false positive results and overestimated effect sizes, and lowers the replicability of published results." I agree with the first part of the sentence, but not with the second (i.e. "…lowers the replicability of published results"). Incomplete reporting and not sharing study materials lowers the ability for others to replicate a study, not use of 'researcher degrees of freedom'. But perhaps I am interpreting "replicability" differently to the authors (given the definition varies so much!) 

Abstract: The authors claim that "Results of comparing random samples of 53 preregistrations from each format indicate that neither format restricted all researcher degrees of freedom and

the Prereg Challenge format performed better on restricting degrees of freedom than the "Standard" format." The latter half of the sentence (re the Prereg Challenge format performing better) is not very clear (as a reader am I to interpret that this format was better for ALL researcher degrees of freedom, and how much better?). It also appears inconsistent with how you interpret the results in the Discussion.

Abstract: The authors claim that "We also found a very low concordance rate among coders about the number of hypotheses (about 15%), suggesting that preregistration is difficult and an acquired skill…" Sorry but I don't follow the logic here. Low concordance among raters appraising preregistrations implies that it is difficult for readers to understand the content of preregistrations or that the coders needed more training, not that that the act of preregistration itself is difficult, in my view. Perhaps clarification on this point would be useful.

Introduction: The authors state that "Specification of the design and the analysis plan before observing the outcomes prevents the outcomes from affecting design and analysis decisions." However, no empirical evidence to support this claim is provided, despite there being a vast literature on reporting biases in medicine that has examined this issue directly that should be acknowledged. Also the paragraph starting, "The potential value of preregistration has been known for a long time…" is also rather limited in my view as it makes no mention of the vast literature on the potential value and observed limitations of registration of clinical trials in medicine. Why not build on that past literature to frame your arguments in the current study? For example, we have known for many years via meta-research studies in medicine that basic registration isn't enough to prevent researcher degrees of freedom, and that detailed protocols and statistical analysis plans are essential (for which a reporting guideline by Gamble in JAMA exists). I realise you may want to focus only on psychology, but I just think that reflecting on this literature could make the rationale for the study stronger.

FOLLOW-UP COMMENT: Oh, I see you refer to this evidence in the Discussion. Hmm, seems a bit odd not to allude to this evidence in the Introduction. 

Methods - Variables of interest: I wonder if an alternative to "Restriction Score" could be used. It has a pretty negative connotation I think, with a possible interpretation of this study being that "preregistration restricts researchers!". Why not something more positive like a "Transparency Score", thus leading to the more positive conclusion that "preregistration improves transparency!".

Results: Minor quibble - there are too many abbreviations in the Results section (e.g. DF, PCR, SPR, DV, IV) which makes it hard to follow. A case in point is the following sentence, which I find dizzying to read: "Higher scores were obtained by PCR than by SPR on DFs that pertained to the operationalization of the variables in the study: D1 ('multiple manipulated IVs', D = 0.22), D3 and A5 ('multiple measures DV' and 'select DV measure', D = 0.19), and D5 and A12 ('additional IVs exclusion' and 'in/exclusion criteria', D = 0.24).". I see no problem with spelling everything out in full. Also, for a medical researcher like me, PCR means something else entirely! 

Table 2. I'm sure this is reported elsewhere but from just reading the table it's unclear to me what the range of possible scores is in the Mean (SD) columns (e.g. is a mean of 1.98 large or small?). A footnote would help 

--

Reviewer #2 (Ulf Toelch, QUEST Center for Transforming Biomedical Research): 

The study by Vanderkamp et al. addresses an important and timely subject namely how current practices in preregistration influence the depth and detail of preregistrations. I also appreciate the openness and transparency to publish the previous reviewer comments. The material on the OSF repository seems complete, I did however not review the code. The manuscript is not an easy read and could still need some editing towards readability. Particularly some procedures are only clear if you read the R code or the supplementary coding scheme. Beyond this I have some major concerns particularly about the analysis that I would like to see clarified.

1. This is reflecting on a point one of the previous reviewers as to the transition from preregistration

to manuscript. A recent preprint at least hints at the possibility that adherence to protocols may not be optimal (10.31234/osf.io/d8wex). It is thus even more important to have high standards for preregistration. It will be harder to adhere to a preregistration with full analysis plan than to a preregistration stating only a hypothesis. It is, however, a lot more work for reviewers to connect a preregistration to a manuscript if the preregistration is long and exhaustive. I feel that the last part of the discussion puts little emphasis on the fact that a. writing good preregistrations is hard b. reviewing them puts a considerable additional burden on reviewers. A more balanced view on preregistration would be warranted and cases where preregistration may not be applicable could be stated.

2. The list of DFs by Wicherts et al. is at least partly targeted specifically at psychology. As PLOS Biology has a broad readership it will be interesting from which fields the preregistrations stem. My guess is that preregistrations will be mainly from psychology. This could be another limitations as e.g. animal experiments may need a different focus for preregistration (https://www.animalstudyregistry.org)

3. I do not find the analysis very compelling. 

 A. The power analysis is done on a Cohens d of .5 You state that as you do not have any prior knowledge about the ES. One than must assume that you think that .5 will be a meaningful effect size. This corresponds to a Cliffs d of .33. This means that only effect sizes equal/larger should be considered as meaningful difference (not only statistically significance). How will smaller effect sizes make a difference between the two preregistration schemes? And if they do why did you not power for them? See my summary point below also. 

 B. It is not clear whether power calculation was conducted on two or one sided test (perhaps I did not read this properly). Your main results is one sided (summary restriction scores) the single DFs are two sided. This needs clarification. 

 C. Your main result (summary restriction scores) is calculated on the mean of median scores. This is not reflected in the power analysis. Some DFs can only have values of 0, 2, and 3, some can have values of 0, 1, 2, 3. As this is your main result your choice to first calculate the median and then the mean needs more motivation. Perhaps you could conduct a non parametric bootstrap to confirm your result. It is also not clear to me what the calculated effect size here means. What would be a viable interpretation?

 D. Table 2. Your data has a clear hierarchical structure (different categories) with a lot of depended data points (within study). This is not reflected in your simple multiple test two group comparison scheme. The preregistered analysis is simplistic and neglects a lot of these effects. What are correlations within study? Will a preregistration that is underreporting in one area also do this in another (as additional material)? 

Additionally you state: "We then conducted exploratory follow-up analyses to investigate on which DFs the two preregistration types differed." in Material and Methods

And: "As preregistered, we conducted follow-up analyses to examine on which of the DFs there were statistically significant differences in the distribution of scores between SPR and PCR." in Results

Does this mean that the whole Table 2 is exploratory and not confirmatory? But is it preregistered? This could be an error in my reading but as it is this is confusing and does not allow a critical appraisal of the results in Table 2. The results also showcase the pitfalls of prespecifying analyses where the data structure is complicated. If I prespecify a suboptimal analysis scheme in a preregistration and reviewers want significant changes will this then result in entirely exploratory analyses in the final paper? I am unsure about the last point myself but want to mention it here to stimulate discussion.

To summarise, I would have wished for a result reporting and discussion based on the effect sizes. What does it mean if there is a difference between the two scores? What is the consequence of such a difference for the quality of the pre-registration. For ordinal variables an argumentation based on the mean values should be avoided. I find the percentages per ordinal level quite illustrative and a Figure for cases like randomisation and missing data, etc. could be helpful. In this context I miss the connection to survey data on QRPs. It would benefit the paper if you could relate the base rates of QRPs that you find for preregistrations to findings from the literature on self reported QRPs (Loewenstein and others). This would also address some of the concerns of previous reviewers and contextualise your results better. I know that space is limited but you could point out where preregistrations are particularly successful and where there is additional effort necessary.

I sign my reviews

Ulf Toelch, QUEST Center for Transforming Biomedical Research

---

## [Decision Letter · Decision Letter 2]

15 Sep 2020

Dear Dr Bakker,

Thank you for submitting your revised Meta-Research Article entitled "Ensuring the quality and specificity of preregistrations" for publication in PLOS Biology. I have now obtained advice from the original reviewers and have discussed their comments with the Academic Editor. 

We're delighted to let you know that we're now editorially satisfied with your manuscript. However before we can formally accept your paper and consider it "in press", we also need to ensure that your article conforms to our guidelines. A member of our team will be in touch shortly with a set of requests. As we can't proceed until these requirements are met, your swift response will help prevent delays to publication. Please also make sure to address the data and other policy-related requests noted at the end of this email.

- a cover letter that should detail your responses to any editorial requests, if applicable

*Copyediting*

*Published Peer Review History*

*Early Version*

Sincerely,

Roli Roberts

Senior Editor,

rroberts@plos.org,

PLOS Biology

DATA POLICY:

Regardless of the method selected, please ensure that you provide the individual numerical values that underlie the summary data displayed in the following figure panels as they are essential for readers to assess your analysis and to reproduce it: Fig 1. NOTE: the numerical data provided should include all replicates AND the way in which the plotted mean and errors were derived (it should not present only the mean/average values).

REVIEWERS' COMMENTS:

Reviewer #1:

[identifies himself as Matthew J Page]

The authors have done an excellent job addressing the reviewers' comments and I have no further suggestions

Reviewer #2:

[identifies himself as Ulf Toelch, QUEST Center for Transforming Biomedical Research]

The authors have dealt with all comments in a detailed and thorough manner. Ambiguities particularly in the analysis were clarified and valuable additional analyses have been conducted. This clearly added to to the transparency of the manuscript. Together with the open code and data it will allow readers to critically assess the evidence presented in the manuscript. Importantly, additional limitations have been pointed out in the discussion. Overall, I think this manuscript will be of great interest to the scholarly community and definitively provoke fruitful discussions.

---

## [Editor Report · Decision Letter 3]

23 Oct 2020

Dear Dr Bakker,

On behalf of my colleagues and the Academic Editor, Lisa Bero, I am pleased to inform you that we will be delighted to publish your Meta-Research Article in PLOS Biology. 

PRODUCTION PROCESS

Before publication you will see the copyedited word document (within 5 business days) and a PDF proof shortly after that. The copyeditor will be in touch shortly before sending you the copyedited Word document. We will make some revisions at copyediting stage to conform to our general style, and for clarification. When you receive this version you should check and revise it very carefully, including figures, tables, references, and supporting information, because corrections at the next stage (proofs) will be strictly limited to (1) errors in author names or affiliations, (2) errors of scientific fact that would cause misunderstandings to readers, and (3) printer's (introduced) errors. Please return the copyedited file within 2 business days in order to ensure timely delivery of the PDF proof. 

If you are likely to be away when either this document or the proof is sent, please ensure we have contact information of a second person, as we will need you to respond quickly at each point. Given the disruptions resulting from the ongoing COVID-19 pandemic, there may be delays in the production process. We apologise in advance for any inconvenience caused and will do our best to minimize impact as far as possible.

EARLY VERSION

PRESS 

Kind regards,

Alice Musson

Publishing Editor, 

PLOS Biology

on behalf of

Roland Roberts,

Senior Editor

PLOS Biology